# Seaweed Inclusion in Finishing Lamb Diet Promotes Changes in Micronutrient Content and Flavour-Related Compounds of Raw Meat and Dry-Cured Leg (Fenalår)

**DOI:** 10.3390/foods11071043

**Published:** 2022-04-04

**Authors:** Vladana Grabež, Elena Coll-Brasas, Elena Fulladosa, Elin Hallenstvedt, Torunn Thauland Håseth, Margareth Øverland, Per Berg, Alemayehu Kidane, Bjørg Egelandsdal

**Affiliations:** 1Faculty of Chemistry, Biotechnology and Food Science, Norwegian University of Life Sciences, NO-1430 Ås, Norway; bjorg.egelandsdal@nmbu.no; 2IRTA, Food Technology and Food Safety Programs, Finca Camps i Armet, E-17121 Monells, Spain; ecollbrasas@gmail.com (E.C.-B.); elena.fulladosa@irta.cat (E.F.); 3Nortura SA, Økern, NO-0513 Oslo, Norway; elin.hallenstvedt@nortura.no (E.H.); per.berg@nortura.no (P.B.); 4Animalia, Økern, NO-0513 Oslo, Norway; torunn.haseth@animalia.no; 5Faculty of Bioscience, Norwegian University of Life Sciences, NO-1432 Ås, Norway; margareth.overland@nmbu.no (M.Ø.); alemayehu.sagaye@nmbu.no (A.K.)

**Keywords:** seaweed, lamb, dry-cured ham, micronutrients, flavour

## Abstract

Innovative feeding strategies tend to improve the quality properties of raw material and dry-cured products. In the present study, Norwegian White female lambs (*n* = 24) were finished during 35 days on three different diets: control (CD), control supplemented with seaweed (5% DM) (SD), and pasture (PD). The quality of raw meat (*Semimembranosus* + *Adductor*) and deboned dry-cured lamb leg (fenalår; *n* = 24) was studied. The heme, SFA, MUFA, and PUFA content in raw meat was not affected with finishing diet. The SD significantly increased the selenium, iodine, and arsenic content in raw meat and in the dry-cured leg the iodine and arsenic. The dry-cured leg from SD-lamb had the highest amount of iodine with 130 µg I/100 g which corresponds to 60% of Adequate Intake. Aldehydes, ketones, and esters in raw meat and dry-cured lamb leg were significantly affected by finishing diet; CD showed increased esters in raw meat and aldehydes in the dry-cured leg compared to SD and PD. The significantly higher content of simple sugars, mannose being the most dominant, was found in the dry-cured leg from SD-lamb compared to CD and PD. Finishing diets had no effect on the taste profile of dry-cured lamb leg. This study showed the potential of seaweed in iodine biofortification of lamb meat and dry-cured products. Iodine-rich meat products should reduce iodine-deficiency among humans.

## 1. Introduction

Fenalår is a dry-cured leg that includes the rump muscles of lamb and sheep. The traditional Norwegian dry-cured meat product “Fenalår fra Norge” is awarded the Protected Geographical Indication [1]. The traditional production of fenalår implies salting and curing of bone-in lamb or sheep leg, while nowadays, industrial production is more oriented towards deboning and netting the legs before salting [2]. A deboned fenalår is an easy-to-slice product and the slices are packed in consumer-friendly packages with standardized weight and shape [2]. 

The commercial value of the dry-cured lamb products may presumably increase with the improved nutritional value. In this sense, different feeding practices have been previously suggested as a strategy to optimize the nutritional value of the raw material used for dry-cured meat production in terms of fatty acid composition [3]. New feeding strategies, such as the inclusion of specific nutrients in finishing animal diets ensure enrichment of raw materials with nutrients important for human health. Supplementation of growing-finishing bull diet with selected nutrients, i.e., selenium, vitamins D_3_, K_3_, and E, increased their contents in the meat [4]. 

The coastal regions of North Atlantic Europe have a long tradition of using seaweeds as a valuable feed for ruminants due to feed shortage. During the last decade, Norway has revealed a renewed interest in seaweed cultivation, with an estimated 20 million tons of harvested seaweed by 2050 [5]. Locally produced seaweed biomass from the cultivation of *Saccharina latissima* could potentially be used as an alternative feed source rich in minerals. Seaweed supplemented diet has shown an increased oxidative stability [6] and an increase of iodine, selenium, and arsenic content in lamb meat [7], thus, increase of iodine and selenium content in pork meat [8]. In addition, consumers identified changes in flavour properties of cooked meatballs when finishing lamb diet was supplemented with 5% seaweed on dry matter base [6]. Therefore, similar effects can be expected in dry-cured products, although, no studies have reported the effect of seaweed supplemented diets on dry-cured products’ quality. Therefore, this study aimed to investigate the effect of finishing lamb diet supplemented with seaweed on volatile compounds and micronutrient content in raw meat and dry-cured leg. Metabolite content, sensory profiles, and labelling advantages of dry-cured leg produced from lamb fed different finishing diets were also analysed.

## 2. Materials and Methods

### 2.1. Animals Selection

Twenty-four Norwegian White female lambs (37.3 ± 1.6 kg BW) about 6 months old were randomly divided in three groups and fed ad-libitum with finishing diets for 35 days: (1) control diet (CD)—total mixed ration of grass silage totally mixed with compound feed, rolled barley and mineral premix, (2) seaweed supplemented diet (SD)–5% of the control diet was replaced with dried *Saccharina latissima* on DM basis, and (3) permanent ley pasture (PD)–for the whole experimental period animals were kept on pasture. Ingredient composition of diets was provided in Table 1. The finishing CD and SD diets were provided in weighed amounts at 0800 h (~40% of the daily allowance) and at 1400 h (the remaining 60% of the daily allowance) in individual pens. All lambs had free access to clean drinking water. Animal procedures for the indoor-fed lambs were approved by the rules and regulations governing animal experiments in Norway under the surveillance of the Norwegian Food Safety Authority (FOTSID:16406). 

All lambs were slaughtered on the same day at a commercial slaughterhouse (Rudshøgda, Nortura SA, Norway) by electrical stunning (3.2 s at 1.3 A), exsanguination followed by electrical stimulation (at 90–100 V AC for minimum 60 sec) and evisceration. Carcasses were hanged separately on hooks and chilled at ~2 °C for 24 h. 

### 2.2. Raw Material 

Lamb legs were cut from CD and SD carcasses 48 h post-mortem and 72 h post-mortem from PD carcasses, subsequently deboned, and connective tissue and subcutaneous fat were removed. A piece of lamb meat composed of the two muscles *Semimembranosus* and *Adductor* (SM+ADD) was dissected from the leg for raw meat analysis. The ultimate pH was measured in SM using a pH meter (Knicks Portamess 913, Berlin, Germany) equipped with Hamilton double pore glass electrode. The raw meat (SM+ADD) was homogenized, vacuum packaged, and stored at −80 °C for fatty acids, heme, micronutrient, and volatile compound analysis.

The remining leg muscles, without SM+ADD, were vacuum packaged and stored at −20 °C for 7 days before dry-cured leg production, followed by analysis of colour, micronutrients, volatiles, metabolites, and sensory properties of the final product.

### 2.3. Production of Dry-Cured Leg

All raw deboned legs were thawed at 15 ℃ for 24 h and processed the same day. Standard salting procedure for deboned fenalår production was used [2] with some modifications; i.e., due to lack of raw material, raw ham from only one leg without SM+ADD was used in a production. In brief, each raw leg was manually rubbed using fine salt, netted and individually salted (4.8 g of salt/100 g raw meat and 144 ppm nitrite) in vacuum shrink bags (polyamide/EVO/polyethylene; oxygen permeability of 12 cc/m^2^/24 h at 23 °C, 0% RH and 1 atm and a water permeability of 8 g/m^2^/24 h at 38 °C, 90% RH and 1 atm; Bemis^®^ Company Inc., Sheboygan Falls, USA). Salted legs were left for a cold phase at 2–4 °C for 42 days. Once the cold face was finished, the legs were treated with a 4% potassium-sorbate solution during 1.5 h at 13 °C, followed by a drying step at 18 °C with a relative humidity (RH) of 60% for 2 days. Afterwards they were dried at 13 °C and RH of 74%, smoked with friction-smoke from beech wood for 6 h, and pressed. The legs were hanged up again at 13 °C and a RH of 74% until achieving a weight loss of ~36%. 

At the end of the process, the sampling of dry-cured leg was performed as follows: a five cm thick slice was obtained from the central part and used for instrumental colour determinations. Then, twenty-four 1.2 mm thick slices were sampled for sensory analyses. Finally, the remaining part of dry-cured meat was trimmed from fat and used for the analysis. The samples were selected randomly from the remaining slice. All the samples were vacuum packed and stored at −80 °C until the analyses were performed. 

### 2.4. Instrumental Colour

A colorimeter Minolta Spectrophotometer CM 700d (Konica Minolta Optics Inc., Osaka, Japan) was used to measure colour in the CIE–LAB space (lightness (L*), redness (a*) and yellowness (b*)) [9] on the 5 cm thick pieces of dry-cured leg. The illuminate used was D65 with 2°. The colour was measured in triplicate on the surface of the dry-cured meat slice in two different areas using closed cone equipment, thus, avoiding zones with fat or connective tissue. The instrument was calibrated according to the manual.

### 2.5. Chemical Analysis

Extraction of fatty acid methyl esters (FAME) from 0.25 g homogenized raw meat was performed as described by Yi et al. [10]. Briefly, homogenized meat sample was mixed with 1 mL of tridecanoic acid (0.5 mg C13:0/mL methanol) as an internal standard. Then, the sample was dissolved and hydrolysed with 0.56 mL of 10 N KOH in water and 4.2 mL of methanol. The incubation of the samples in water bath at 55 °C for 1.5 h with hand shaking for 5 s every 20 min was performed. The samples were cooled, 0.46 mL of 24 N sulphuric acid in water was added, then once again incubated and cooled as previously described. Mixing with 3 mL of hexane for 5 min and centrifuging at 653× *g* for 10 min the fatty acid methyl esters (FAME) were separated. The hexane layer with FAME was transferred to GC vials and kept at −20 °C prior to the analysis. The FAME analyses were carried out on a Carlo Erba GC 8000 GC instrument equipped with a Carlo Erba EL 980 Automatic Sampler, a flame ionization detector (Carlo Erba AS V570 FID, Carlo Erba Instruments, Milano, Italy) and a CP 88 capillary column (length: 50 m, i.d.: 0.25 mm, film thickness: 0.20 µm; Varian, Agilent Technologies, Matriks, Norway) [11]. The FAME present in the samples were identified and quantified using four standard solutions prepared from a FAME Mix (Supelco 37 component FAME Mix). The concentration of the fatty acids (FA) was expressed as mg FA/100 g of meat.

Heme analysis was performed on homogenized raw meat [12]. The meat sample (0.155 g) was mixed with distilled water, acetone and HCl acid (37%), and then centrifuged at 16,000× *g* for 10 min. The absorbance of supernatant was measured on a spectrophotometer at 407 nm (Gen 5). All the assays were performed in duplicates as a minimum, preferably triplicates. Prior to the analyses, a standard curve using different concentration of myoglobin was performed and the absorbance was measured at 525 nm using Myoglobin (mM) = A (525 nm)/(7.6 mM^−^^1^ cm^−^^1^ * 1 cm).

The micronutrient analyses were performed on raw meat and dry-cured leg. For the selenium (Se) and arsenic (As) analysis, freeze-dried sample (0.25 g) was subjected to digestion with 5.0 mL of Ultrapure HNO_3_ and 2 mL of MilliQ water. Selenium (Se) (99.9% purity) was added as internal standard (4 µg/L in the final dilution). The sample was digested at 260 °C degrees in an UltraClave from Milestone. After the digestion, sample was diluted to 50 mL with MilliQ water and1 mL of C_4_H_13_NO. With constant shaking, the sample was heat treated overnight at 60 °Cand additionally for 1 h at 90 °C. The sample was then analysed with an Agilent 8800 ICPMS in Oxygen reaction mode with quadrupole for both Se and Arsenic (As). Se (corrected for natural Se) was used as internal standard for both As and Se. 

The iodine analyses were performed in the accredited laboratory of National Institute of Health Doutor Ricardo Jorge (INSA, Lisboa, Portugal) [13]. Sample (1 g) was weight into a 50 mL Falcon tube and extracted by graphite block system (DigiPREP, SCP Science, Courtaboeuf, France) with TMAH (tetramethylammonium hydroxide) at 90 °C for 3 h. After that sample was centrifuged and filtered through a 0.45 μm filter. Iodine was determined by using an inductively coupled plasma mass spectrometry (ICP-MS, Thermo X series II) equipped with autosampler Cetac ASX-520. The iodine content quantification was performed using an iodine calibration curve with the correlation coefficient ≥ 0.9995. 

### 2.6. Volatile Compound Analysis

The volatile compounds (VOC) were isolated from 5 g of homogenized raw meat and dry-cured leg cut in cubes (2 × 2 × 2 mm) weight in an aluminium cylindrical container (50 mL) and set in Micro-Chamber sample pot. As internal standard heptanoic acid ethyl ester (0.04 mg/mL of methanol) was used. The headspace volatiles were extracted on a Micro-Chamber M-CTE250 (Markes, Bridgent, UK) using the Tenax tubes (SS con TD tubes, hydrophobic Tenax TA/Carbograph 1TD, Markes International Ltd., Bridgent, UK) at moderate temperature of chamber 30 °C and a flow of 50 mL N_2_/min during 60 min. Water from the Tenax tubes was removed with a nitrogen (50 mL N_2_/min), in the opposite direction to the sampling. The flow was controlled with a universal gas flowmeter (ADM1000, Agilent, Santa Clara, CA, USA). Volatiles trapped on Tenax were desorbed in a Perkin Elmer Automatic Thermal Desorption System ATD400 at 250 °C for 5 min and transferred to a Gas Chromatograph 6890N (Agilent Technologies, Santa Clara, CA, USA) equipped with DB–WAXetr fused silica capillary column (30 m × 0.25 mm i.d., 0.5 µm film thickness; J&W Scientific, Folsom, CA, USA) and an ion source Agilent 5975. The carrier gas was He (99.9999%) with a flow rate of 1.0 mL/min. The GC temperature was: 30 °C for 10 min, increased 1 °C/min to 40 °C, 3 °C/min to 70 °C, 6.5 °C/min to 160 °C, and 20 °C/min to 230 °C with a final hold time of 4 min. Analysis time was 51.35 min and recorded mass range was m/z 33−300. Volatile analyses were performed in duplicate.

The mixture of Miglyol 812 (AXO INDUSTRY, Warve, Belgium) and standards of butanal (99%), cis-2-penten-1-ol (95%), 2-undecanone (99%), dimethyl sulfone (98%), hexanal (98%), phenol (99.5%), octanoic acid, methyl ester (>98%) (Sigma-Aldrich Chemie GmbH, Schnelldorf, Germany), and acetic acid (100%, VWR, Fontenay-saus-Bois, France) was run as a control in a sequence. 

Deconvolution and integration of peaks were performed with MassHunter Qual (version B.07.00, Agilent Technologies, Santa Clara, CA, USA), thus identification using NIST17 (National Institute of Standards and Technology, Gaithersburg, MD, USA) with ≥70% mass spectral match. The compound alignment was carried out with Qual add-in tool SearchReview (Leoson BV, Middelburg, The Netherlands). VOC present in ≥50% of the analysed samples from one group were kept for further analysis. Semi-quantification of identified VOCs was carried out using calibration curve of the standards mixed in Miglyol. 

### 2.7. Metabolite Analysis

Metabolite extraction, derivatization and analysis were performed on homogenized dry-cured leg [14]. In brief, the extraction of metabolites from 1 g of dry-cured leg sample was carried out with a water: methanol: chloroform mixture and ribitol was used as internal standard. After incubation, sonication and centrifugation 1.5 mL of the supernatant were transferred in spin-x tubes and centrifuged for 3 min at 1500× *g* to remove the salt. Then, 1 mL of desalted supernatant was dried in a SpeedVac at room temperature. The dried sample was dissolved and derivatized. The analysis was performed on GC−MS (1310-ISQ QD single quadrupole GC−MS instrument from Thermo Fisher, Waltham, MA, USA) equipped with a capillary column (CP9012 VF-5 ms 30 m, ID 0.25 mm and 0.25 µm film thickness with 5 m EZ-Guard, Agilent). Metabolite analyses were performed in duplicate.

Deconvolution and integration of peaks were performed with MassHunter Qual (version B.07.00, Agilent Technologies, Santa Clara, CA, USA), and identification using NIST17 (National Institute of Standards and Technology, Gaithersburg, MD, USA) with ≥ 70% mass spectral match. Compound alignment was carried out with Qual add-in tool SearchReview (Leoson BV, Middelburg, The Netherlands). Mass spectra of identified compounds were confirmed using GOLM metabolome database (Max-Planck Institute for Molecular Plant Physiology, Golm, Germany). Only compounds present in ≥50% of analysed samples from one group were kept for further analyses. 

The internal standard was used to normalize the area of identified compounds when the internal standard was more than one std dev different. An alkane standard mix (C8–C40) was used to calculate retention index (RI) of identified compounds. The semi-quantification of identified metabolites was carried out using calibration curve of the standards: glycerol (85%), glucose anhydrous (Merck, Darmstadt, Germany), succinic acid (99.5%), methionine (98%), and myristic acid (≥99%) (Sigma Aldrich, St. Louis, MO, USA). 

### 2.8. Sensory Analysis

Dry-cured legs (*n* = 4) from each dietary group were randomly selected and assessed (*n* = 12) by a semi-trained panel. The internal semi-trained panel is reliable and trained to evaluate sensory properties of dry-cured lamb products. The selected panellists are yearly re-trained for sensitivity of recognition thresholds of basic tastes (sweet, sour, salty, bitter, and umami) and rancidity. The panel consisted of 12 members; 3 males and 9 females. The slices of dry-cured leg (2.1 mm thick) were kept at room temperature for 1 h. The Quantitative Descriptive Analysis (QDA) of overall odour intensity and four taste attributes (sheep, sweet-honey, rancid, and salty) were carried out by using an intensity scale from 1 to 9 (whereas 1 is the lowest incidence of the trait and 9 the highest intensity), and texture attributes (bite through capacity, tenderness, hardness, softness, juiciness, adhesiveness and/or dryness) were carried out by using an intensity scale from 1 to 5 (whereas 1 is the lowest incidence of the trait and 5 the highest intensity). The sensory analysis was carried out as follows: all samples were chewed, both core and surface, and the assessors made an average of each trait. All samples were evaluated by the panellists with a different order and each sample received a 3-number code randomly selected. Between all the samples, water and cucumber were used to rinse the mouth, and a 5 min break was carried out after the evaluation of five samples. One sample was replicated among all assessors.

### 2.9. Statistical Analysis

The effect of finishing diet on pH, heme, fatty acids, and micronutrient content in raw meat (*n* = 24), micronutrients, metabolite content, and sensory scores in dry-cured leg (*n* = 24), was determined by one-way ANOVA (Minitab 18, Minitab Ltd., Coventry, UK). 

Both metabolite and volatile results were handled as follows. The samples were tested for normality based on Anderson-Darling test for normality and Grubbs’ Test for outliers (both assessed at significance level *p* < 0.05). This approach effectuated that one sample was removed for 59% of VOC calculations. Thus, only VOC with more than four carbons were regarded of interest for further analysis. 

For volatiles, the effects of dietary treatment and dry-cured processing were tested using General Linear Model (GLM) analysis (Minitab 18, Minitab Ltd., Coventry, UK). The metabolites were first analysed using the Mass Profiler Professional 15.0 (Agilent Technologies). The identified metabolites transformed into normalized area spectra were log2 transformed and a Fold change analysis (FC > 2.0), applying Benjamini-Hochberg multiple test corrections, was performed. The effect of diet on metabolite changes was analysed using One way ANOVA, since dry-curing was not a design variable. Post hoc tests were carried out using either Turkey or Games Howell methods pending equal or inequal variance (Minitab 18, Minitab Ltd., Coventry, United Kingdom). 

PLS models were built using Unscrambler, version X10.1 software (Camo, Trondheim, Norway). Odour intensity and taste attributes were regressed on selected volatile (C > 4) and identified metabolites. Scores for seven validated factors were tested using one-way ANOVA to investigate diet effect on flavour profile. 

## 3. Results and Discussion

### 3.1. Quality Properties

The Table 2 shows pH values of *Semimembranosus* muscle (SM), heme and fatty acid content in raw meat (SM+ADD) from lambs fed three different diets for 35 days pre-slaughtering. The pH values of the studied SM were significantly (*p* = 0.01) lower in SD but the pH differences were not relevant in practice. Regarding to the heme, SFA, MUFA, and PUFA contents no significant differences (*p* > 0.05) were found in SM+ADD between animals fed the three finishing diets. Only arachidonic acid (C20:4n-6) was significantly (*p* < 0.001) affected by growing-finishing diet, with PD having higher content than CD and SD. Similar results were reported for grass-fed ground beef compared to grain-fed [15]. 

Finishing diet had no significant effect (*p* > 0.05) on heme content and colour (L*, a* and b* values) of dry-cured leg (having mean values of L* = 32.29, a* = 11.38, and b* = 4.15). This is in agreement with previous studies on deboned fenalår [2].

### 3.2. Micronutrient Content 

Seaweed inclusion in finishing lamb diet significantly increased the Se level (Table 3) in raw meat (*p* < 0.001) compared with CD and PD. A similar trend was seen in dry-cured leg, where SD and CD contained significantly higher (*p* < 0.001) level of Se compared to PD. In 100 g of product, the Se content was significantly lower (*p* = 0.003) in SD dry-cured leg than in raw meat. However, the variation in Se content in SD raw meat and dry-cured leg was lower compared with CD, indicating stabilization of Se with seaweed inclusion in finishing lamb diet. 

Based on EFSA’s recommendation, an Adequate Intake (AI) for Se is 70 μg/day for adults [17]. The Se content in 100 g of CD and SD dry-cured leg presents at average 22% and 24% of the AI, respectively. In addition, the average Se content in PD dry-cured leg corresponds to only 17% of the AI. 

The I content significantly increased (*p* < 0.001) in both raw meat and dry-cured leg with inclusion of seaweed in finishing lamb diet (Table 3). An increase of I content a 34-fold in homogenized meat was reported when lamb diet was supplemented with I-rich seaweed (74 mg I/kg feed) [7]. In SD lamb, the I content was significantly higher (*p* < 0.001) in dry-cured meat compared with raw meat. The difference could be due to processing or variations between muscles. Almost the double I content in *Gluteus medius* when compared to *Longissimus thoracis* muscle was found when beef cattle was fed a I-supplemented diet [18]. Thus, significant differences in I content between individual muscles (*p* < 0.05) was reported; *Trapezius* and *Biceps brachii* muscles had higher I level compared to *Longissimus*, *Psoas major*, and *Semimembranosus* when the bovine diet was supplemented with 400 mg ethylenediamine dihydroiodide and fed over four weeks [19]. 

The iodine AI for adults is 150 μg I/day [17]. Considering minimum content of I in SD dry-cured leg, 100 g of this product contains amount of I that corresponds to 60% of AI. In addition, the I content in 25 g of SD dry-cured leg is equal to 20 g of the I fortified salt.

The As content in raw meat and dry-cured leg was significantly higher (*p* < 0.001) in SD lamb compared with CD and PD (Table 3). The content was significantly higher (*p =* 0.048) in dry-cured leg compared with raw meat; however, if this is an effect of processing or variation between muscles is not known. The As is considered a contaminant in food, more toxic when present in inorganic form. However, over 90% of As in seaweed is presents in the organic form [20]. The maximum allowed content of inorganic As in rice and rice products is in the range between 100 and 300 µg/kg [21]. Total content of As found in raw meat and dry-cured leg produced from SD lamb was 3.8 and 4.5 µg/100 g, respectively. In general, As content in dry-cured leg is well below the limits for rice. 

### 3.3. Volatile Compounds

The volatile analyses were employed to characterize changes of flavour-related compounds in raw meat (SM+ADD) and dry-cured leg. Fifty-five volatiles were detected, however, only 12 of the identified volatile compounds (VOCs) were significantly affected by finishing diet, processing, or their interaction effect (*p* < 0.05, Table 4). Processing and its interaction with diet most frequently had a significant effect (*p* < 0.05), indicating complexity of development of flavour-related compounds. Since most volatiles were similar, odour may only to a limited extent be diet-dependent for lamb meat.

The total content of alkanes increased numerically in dry-cured leg compared with raw meat in CD and SD groups. The opposite trend was found in the PD group, with decrease of total alkanes in the dry-cured leg (the interaction effect being *p* = 0.012). An increase of alkanes can be partially attributed to lipid oxidation during dry-cured meat production [22]. The reduction of alkane content in PD dry-cured leg was most likely related to improved meat oxidative stability of pasture fed animals. Undecane was identified in dry-cured meat at a level above odour threshold. However, alkanes have generally been considered as non-contributors to meat flavour [23] with low odour activity in dry-cured meat [24].

Aldehydes, ketones and alcohols are compounds that typically originate from the degradation of fatty acids. Normally, lipid oxidation during the dry-curing process occurs, first the PUFA-rich phospholipids then the triglycerides oxidize [25]. The total amount of aldehydes was higher in CD dry-cured leg compared with raw meat. However, these compounds showed numerically reduced total content in PD and SD dry-cured leg compared with raw meat. Thus, the contents of heptanal and nonanal, being degradation products of C18:2n-6 and C18:1n-9, were numerically lowest in the PD group. This could either mean less degradation of unsaturated fatty acids or an accelerated reaction of formed aldehydes with NH_2_-group or other reactive groups [26]. The total level of alcohols and ketones in SD and CD dry-cured legs was similar, with PD dry-cured leg values being numerically or significantly lower. Since SD dry-cured leg had lower values for total aldehydes, total alcohol and some ketones relative to CD, also the SD may contribute with antioxidants. Some aldehydes, ketones and alcohols are in the range where they could affect the odour of the dry-cured leg. However, it is not possible to indicate from lipid degradation components how the consumers will assess the flavour of dry-cured product.

The total level of esters decreased in all dry-cured legs, while decanoic acid, methyl ester clearly increased compared to raw meat. Sirtori et al. [27] reported that 15 out of 16 identified esters in pork ham were ethyl esters with decanoic acid, methyl ester as an exception. Esters are formed from lipid degradation compounds that are either acids or alcohols. Microbial esterases can also produce esters [28], therefore some esters in the present study may originate from microbial activity, while others are lipid oxidation products, i.e., decanoic acid, methyl ester. SD dry-cured leg showed numerically higher content of the two esters having fruity flavour, indicating a positive effect of seaweed-supplemented finishing lamb diet.

**Table 4 foods-11-01043-t004:** Differences in volatile compounds (*p* < 0.05) between raw meat and dry-cured leg from lamb fed three finishing diets. Results are presented as mean ± STD (*n* = 8).

Compound ^1^	RT	CD ^2^		SD		PD		*p*-Value			Threshold (mg/kg)	Odour Description
	(Min)	Meat ^3^	Dry-Cured Meat	Meat	Dry-Cured Meat	Meat	Dry-Cured Meat	D ^4^	P	D × P		
undecane	22.3	<0.001 ^b^	2.42 ± 1.13 ^a^	<0.001 ^b^	2.69 ± 1.75 ^a^	<0.001 ^b^	0.25 ± 0.01 ^ab^	0.200	<0.001	0.200	0.4–3.26 ^(1)^	alkane ^(2)^
heptanal	33.3	0.31 ± 0.24 ^ab^	0.47 ± 0.32 ^a^	0.31 ± 019 ^ab^	0.37 ± 0.23 ^ab^	0.34 ± 0.26 ^ab^	0.005 ± 0.00 ^b^	0.233	0.028	<0.001	0.3–1.4 ^(2)^	unpleasant fat, (dilute:citrus/nut) ^(2)^
nonanal	38.0	2.40 ± 1.51 ^ab^	10.84 ± 14.16 ^a^	2.33 ± 1.32 ^ab^	1.71 ± 1.10 ^b^	3.32 ± 1.81 ^ab^	0.03 ± 0.01 ^b^	0.039	0.688	0.045	0.4 ^(2)^	unpleasant fat, (dilute:citrus/rose) ^(2)^
2-ethyl-1-hexanol	40.3	0.003 ± 0.00 ^b^	0.29 ± 0.16 ^a^	0.006 ± 0.01 ^b^	0.25 ± 0.19 ^a^	0.01 ± 0.01 ^b^	0.01 ± 0.00 ^b^	0.096	<0.001	0.006	0.14 ^(3)^	unpleasant, fat, citrus, green, oil, rose ^(2)^
3-heptanol	35.0	<0.001 ^b^	0.13 ± 0.11 ^a^	<0.001 ^b^	0.12 ± 0.09 ^a^	<0.001 ^b^	<0.001 ^b^	0.132	<0.001	0.127	0.03 ^(4)^	alcoholic, herb, overripe fruit, pleasant ^(2)^
1-octanol	41.7	<0.001 ^b^	0.44 ± 0.48 ^a^	<0.001 ^b^	0.14 ± 0.10 ^ab^	<0.001 ^b^	0.003 ± 0.00 ^b^	0.038	0.001	0.025	0.023 (air, mg/m^3^) ^(5)^	fatty, green, citrus, fruity, metal ^(2)^
6-methyl-5-hepten-2-one	36.4	0.73 ± 0.34 ^b^	13.04 ± 6.71 ^a^	1.57 ± 0.69 ^b^	9.09 ± 4.45 ^ab^	1.38 ± 052 ^b^	0.14 ± 0.08 ^bc^	0.366	<0.001	0.028	1–8.2 (water, mg/l) ^(6)^	fatty, green, citrus, fruity ^(2)^
acetophenone	44.0	1.10 ± 0.64 ^b^	9.08 ± 6.91 ^a^	1.46 ± 0.86 ^b^	7.95 ± 5.09 ^ab^	0.99 ± 0.75 ^b^	0.14 ± 0.05 ^b^	0.176	0.003	0.023	0.17 ^(7)^	almond, animal, flower, must, plastic ^(2)^
3-nonanone	36.9	0.018 ± 0.00 ^b^	9.40 ± 5.51 ^a^	0.018 ± 0.00 ^b^	7.43 ± 4.15 ^ab^	0.018 ± 0.00 ^b^	0.20 ± 0.07 ^b^	0.822	<0.001	0.193	0.05 ^(8)^	floral ^(9)^
hexanoic acid, methyl ester	30.1	3.15 ± 1.34 ^a^	5.67 ± 3.32 ^a^	3.34 ± 1.43 ^a^	6.06 ± 2.54 ^a^	3.58 ± 1.53 ^a^	0.18 ± 0.07 ^b^	0.027	0.151	0.007	0.07 ^(10)^	fruit, pineapple, sweet ^(2)^
octanoic acid, methyl ester	37.9	2.09 ± 0.82 ^a^	4.52 ± 2.07 ^a^	1.93 ± 0.92 ^a^	4.59 ± 1.58 ^a^	1.67 ± 0.85 ^ab^	0.11 ± 0.02 ^b^	<0.001	0.560	0.005	0.2-0.9 ^(11)^	fruit, orange, sweet, wax, wine ^(2)^
decanoic acid, methyl ester	42.5	0.26 ± 0.15 ^b^	1.48 ± 0.71 ^a^	0.26 ± 0.13 ^b^	3.03 ± 1.18 ^a^	0.15 ± 0.13 ^a^	0.03 ± 0.01 ^b^	<0.001	<0.001	0.003	1.0 ^(12)^	fresh, wine, fruity, fat ^(2)^
Total content ^5^												
alkanes		1.89 ^b^	5.48 ^a^	1.59 ^b^	4.76 ^a^	3.41 ^ab^	1.33 ^b^	0.421	0.065	0.012		
aldehydes		4.8 ^b^	12.4 ^a^	4.7 ^b^	2.4 ^b^	4.8 ^a^	0.85 ^b^	0.040	0.809	0.040		
alcohols		3.2 ^a^	1.49 ^abc^	2.8 ^ab^	0.97 ^bc^	3.2 ^a^	0.59 ^c^	0.507	0.001	0.616		
ketones		127 ^ab^	135 ^a^	52 ^bc^	126 ^ab^	74 ^abc^	27 ^c^	0.017	0.601	0.089		
esters		70 ^a^	26 ^b^	29 ^b^	24 ^b^	28 ^b^	4.5 ^b^	0.003	0.002	0.096		

^a–c^ Different subscripts in the same raw indicate differences (*p* < 0.05) between raw meat and dry-cured leg;.^1^ Volatile compounds were expressed as mg/kg of dry-cured leg; ^2^ Control diet = CD, Seaweed supplemented diet = SD, Pasture diet = PD; ^3^ VOC content in raw meat (*Semimembranosus* + *Adductor*) was corrected for the water loss (~36%) during dry-cured leg production; ^4^ D = diet; P = product; D × P = diet and product interaction; ^5^ Total content of VOCs with > C_4_: 15 alkanes, 5 aldehydes, 9 ketones, 10 alcohols, 9 esters. ^(1)^ [29], ^(2)^ [30], ^(3)^ [31], ^(4)^ [32], ^(5)^ [33], ^(6)^ [34], ^(7)^ [35], ^(8)^ [36], ^(9)^ [37], ^(10)^ [38], ^(11)^ [39], ^(12)^ [40].

### 3.4. Metabolite Changes

Suspected untargeted GC-MS approach identified 56 metabolites in dry-cured leg produced from lamb fed three different finishing diets. Of these, 46 metabolites of dry-cured leg showed the changes (FC > 2) between different diets (Table 5) and 13 were significantly affected (*p* < 0.05) by finishing diet (Table 6). 

Up-regulated free amino acids in SD compared to CD and PD dry-cured leg indicated that protein breakdown process was intensified during meat processing and maturation. Differences in accumulation of free amino acids were explicit when SD and PD dry-cured leg compared. Formation of free amino acids in dry-cured ham depends on endogenous proteases and peptidases [41]. Among them, cathepsin is most stable and active through dry-curing process [42]. However, the activity of proteolytic enzymes (cathepsins and exoprotease) in the muscle of Iberian pig appeared to be reduced with alteration of animal movements [43]. In the present study, dry-cured meat produced from deboned legs of CD and SD lamb reflected up-regulation in individual amino acids, i.e., leucine, β-alanine, serine, phenylalanine, asparagine, tyrosine, compared with outdoor grazing animals (Table 5). Total amino acid (*p* = 0.002) and serine (*p* = 0.018) content was significantly higher in CD and SD dry-cured leg, but tryptophan (*p* = 0.030) was higher in PD (Table 6). Moreover, supplementation of seaweed to indoor fed animals resulted in up-regulation of glycine, leucine, isoleucine, β-alanine, and methionine, but down-regulation of histidine. Mineral enrichment of lamb finishing diet through seaweed supplementation possibly reduced histidine in SD dry-cured leg due to chelating metal ions. 

Free amino acids can have a direct effect on the taste of dry-cured leg, or act as precursors in Maillard and Strecker reactions, yielding other flavour-related compounds [44]. Therefore, the presence of these metabolites, known as sour and bitter compounds, above the taste threshold (Table 6) may have implications on the sweet-honey and sheep taste attributes (Figure 1). 

The up-regulation of free fatty acids in the SD dry-cured leg may have occurred due to increased lipase activity. In general, SFA and MUFA content were numerically higher and PUFA lower in SD raw meat compared with CD and PD. Similar trends were observed in dry-cured leg. Regarding PUFA 9,12-octadecadienoic acid (Z,Z)- (18:2), arachidonic acid, and eicosapentaenoic acid, these were up-regulated in SD dry-cured leg compared with CD indicating higher lipolytic activity and reduced lipid oxidation. 

The results for sugars and sugar-related alcohols (scyllo- and myo-inositol) in dry-cured leg imply higher glycogen storage and *post-mortem* breakdown in indoor fed animals compared with grazing animals. In addition, SD dry-cured leg showed significantly higher total sugar content (*p* < 0.001), mannose (*p* = 0.005), and ß-alanine (*p* = 0.032) as sweet taste compounds than CD and PD. However, other compounds with sour (lactic acid, malic acid, and butanedioic acid) and bitter (creatinine and niacinamide) taste were significantly higher in SD dry-cured leg than other two groups indicating complex underlying processes in taste formation. 

**Table 6 foods-11-01043-t006:** Differences in metabolites (*p* < 0.05) between dry-cured leg produced from lamb fed three finishing diets. Results are presented as mean ± STD (*n* = 8).

Compound ^1^	CD ^2^	SD	PD	*p*-Value	Threshold (mg/kg)	Taste Description
serine	5325 ± 241 ^a^	4375 ± 336 ^ab^	3085 ± 347 ^b^	0.018	0.2 ^(1)^	sweet, umami, sour ^(2)^
tryptophan	292 ± 20 ^b^	405 ± 41 ^ab^	625 ± 75 ^a^	0.030	102 ^(3)^	bitter ^(3)^
mannose	3587 ± 190 ^ab^	4318 ± 137 ^a^	2460 ± 133 ^b^	0.005	~12,000 ^(4)^	sweet ^(5)^
eicosapentaenoic	123 ± 11 ^b^	154 ± 14 ^ab^	260 ± 31 ^a^	0.025	n.d.	n.d.
ß-alanine	168 ± 4 ^b^	319 ± 9 ^a^	171 ± 4 ^b^	0.032	107 ^(6)^	slight sweet ^(7)^
lactic acid	14,737 ± 882 ^ab^	18,311 ± 293 ^a^	14,129 ± 895 ^b^	0.014	10 ^(8)^	sour ^(9)^
4-hydroxybutanoic acid	40.6 ± 3.7 ^b^	80.6 ± 5.5 ^a^	43.2 ± 5.1 ^b^	0.002	n.d.	n.d.
malic acid	58.1 ± 4.1 ^b^	104 ± 7.0 ^a^	87.2 ± 11.3 ^ab^	0.007	9.6–99 ^(10)^	sour, tart ^(7)^
2-aminomalonic acid	56.9 ± 4.1 ^b^	136 ± 18 ^a^	99.8 ± 13 ^ab^	0.023	n.d.	n.d.
butanedioic acid	542 ± 42 ^b^	851 ± 40 ^a^	680 ± 70 ^a^	0.038	0.2 ^(8)^	sour ^(9)^
creatinine	610 ± 67 ^ab^	1021 ± 119 ^a^	348 ± 42 ^b^	0.009	n.d.	bitter ^(11)^
ethanolamine	96 ± 19 ^b^	615 ± 119 ^a^	449 ± 45 ^ab^	<0.001	~2 ^(12)^	weak ammonia ^(12)^
niacinamide	80 ± 7.0 ^b^	130 ± 7.9 ^a^	110 ± 14 ^ab^	0.035	n.d.	bitter ^(13)^
Total content ^3^						
amino acids	51,544 ^a^	51,667 ^a^	43,453 ^b^	0.002		
fatty acids	3552 ^c^	3254 ^b^	4609 ^a^	0.026		
sugars	8190 ^b^	9386 ^a^	6033 ^c^	<0.001		

^a–c^ Different subscripts in the same raw indicate differences (*p* < 0.05) between dry-cured legs; ^1^ Metabolites were expressed as mg/kg of dry-cured leg; ^2^ Control diet = CD, Seaweed supplemented diet = SD, Pasture diet = PD; ^3^ Total content of 17 amino acids, 6 fatty acids, and 6 sugars; ^(1)^ [45], ^(2)^ [46], ^(3)^ [47], ^(4)^ [48], ^(5)^ [49], ^(6)^ [50], ^(7)^ [51], ^(8)^ [30], ^(9)^ [52], ^(10)^ [53], ^(11)^ [54], ^(12)^ [55], ^(13)^ [56], n.d. = not defined.

### 3.5. Sensory Profile of Dry-Cured Leg

In general, the sensory analysis showed no significant effect (*p* > 0.05) of finishing lamb diet on overall odour and taste of dry-cured leg. Odour intensity score was numerically higher for PD dry-cured leg compared with CD and SD. High scores for odour indicate saturation of olfactory senses; scores were as follows PD > CD > SD (8.2, 7.7, and 7.5, respectively). As shown in Figure 1A, dry-cured leg from all groups had similar scores for salty and rancid taste. However, small numerical differences were found for sheep and sweet-honey taste. Sweetness score was reduced for PD dry-cured leg, while the score for sheep taste was higher for SD. 

Selected volatiles (C > 4) and all identified metabolites were related to sensory attributes of dry-cured leg using PLS analysis to explore the chemical drivers of flavour properties (odour and taste). No validated PLS model was obtained (RMSE = 0.51, R^2^ = n.a.), and this may indicate limitations by the small sample size for prediction of sensory attributes. However, fitted models revealed (*p* = 0.021) that the variation in odour and taste profile selected many of the variables measured, thus, supported a complexity of chemistry behind the flavour.

In regards to studied texture attributes of dry-cured leg from lamb fed different finishing diets (Figure 1B), SD dry-cured leg showed a significantly lower scores for bite through (*p* = 0.007) compared with CD and PD. The CD dry-cured leg had significantly higher tenderness (*p* = 0.007) and softness (*p* = 0.014) compared with SD and PD. The results obtained for tenderness could be related with up-regulation of free amino acids in CD and SD dry-cured leg as shown in Table 6. However, no differences (*p* > 0.05) was found for juiciness, hardness, and dryness. 

## 4. Conclusions

The results obtained in this study indicate great potential of seaweed inclusion in finishing lamb diet to enrich raw meat and dry-cured product with iodine and to stabilize the selenium content. In further studies will be necessary to optimize the inclusion level of seaweed and period of finishing feeding for lamb and sheep. The proposed models for evaluation of the interaction among flavour-related compounds and sensory attributes should be tested on greater number of samples. Understanding the effect of finishing diet on flavour profile of the dry-cured product is a key step in further improvement of its flavour quality. 

## Figures and Tables

**Figure 1 foods-11-01043-f001:**
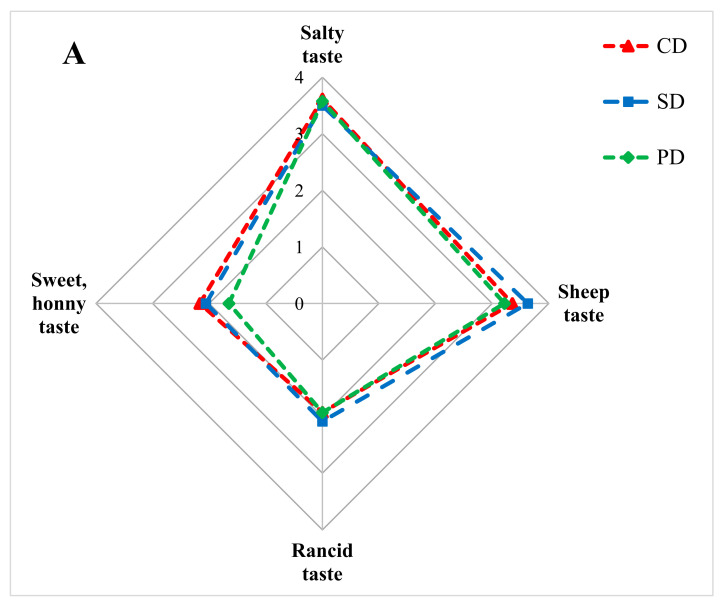
Taste (**A**) and texture (**B**) profiles of dry-cured leg (*n* = 4) produced from lamb fed three different finishing diets (Control diet = CD, Seaweed supplemented diet = SD, Pasture diet = PD). Labelled attributes (*) showed significant difference (*p* < 0.05).

**Table 1 foods-11-01043-t001:** Ingredient (g/kg) and chemical composition of three finishing lamb diets.

Ingredients	CD ^1^	SD	PD ^2^
Early cut grass/clover silage	822.0	822.0	-
Wilted seaweed ^3^	-	67.9	-
Compound feed (DRØV lam) ^4^	102.2	102.2	-
Rolled barley	21.6	-	-
VitaMineral ^®^ Normal Sau ^5^	4.8	4.8	-
GrassAAT Korn ^6^	3.1	3.1	-
Added free water	46.3	-	-
Analyzed content, g/kg DM			
Crude protein	190.3	186.4	181.3
Neutral detergent fiber, NDF	453.5	433.4	552.3
Acid detergent fiber, ADF	282	279.1	253.4
Ash	91.9	105.6	88.8
Organic matter	908.1	894.4	911.2
Starch/WSC ^7^	73.4	61.3	95.6
Analyzed content of minerals, mg/kg DM			
Iodine	5.40	204.60	n.a. ^8^
Selenium	0.40	0.35	0.03
Arsenic	0.14	3.66	n.a.

^1^ Control diet = CD, Seaweed supplemented diet = SD, Pasture diet = PD; ^2^ Grazed outside on a permanent ley-pasture, with access to mineral lick stone for sheep (Pluss Saltstein Hvit, Felleskjøpet, Norway); ^3^ Wilted seaweed with DM content of 283 g/kg, and 118, 373, 234, 409 g/kg DM of CP, aNDFom, aADFom, ash in respective order and gross energy value of 10.7 MJ/kg DM; ^4^ Commercial compound feed produced and supplied by Norgesfôr AS (Mysen, Norway), with CP, aNDFom and crude fat contents of 153, 235, and 43 g/kg, respectively; ^5^ Vitamin and mineral supplement for sheep (Vilomix Norway AS; Hønefoss, Norway) containing vitamins (A 100 IE/g; D3 100 IE/g; and E 2000 mg/kg), macro-minerals (g/kg of Ca 140; P 70; Mg 60; Na 90 and S 10), and micro-minerals (mg/kg of Mn 3000; Zn 5000; Co 30; I 100 and Se 25); ^6^ GrassAAT Korn is a preservative and feed stabilizer; ^7^ Starch/WSC = starch content of the total mixed ration fed indoors and water-soluble carbohydrate for grazed sward; ^8^ n.a.= not analysed in pasture sample.

**Table 2 foods-11-01043-t002:** pH, heme content, and fatty acid profile of raw meat from lamb fed three finishing diets. Results are presented as mean ± STD (*n* = 8).

	CD ^1^	SD	PD	*p*-Value
pH ^2^	5.53 ±0.02 ^ab^	5.47 ± 0.04 ^b^	5.54 ± 0.06 ^a^	0.010
Heme content (mg/g w/w) ^3^	2.68 ± 0.85	2.58 ± 0.49	2.97 ± 1.21	0.675
Fatty acids (mg/g of meat) ^3^				
C10:0	0.06 ± 0.03	0.07 ± 0.04	0.05 ± 0.02	0.285
C12:0	0.10 ± 0.06	0.14 ± 0.09	0.08 ± 0.05	0.301
C14:0	1.15 ± 0.61	1.53 ± 1.00	0.93 ± 0.55	0.289
C15:0	0.19 ± 0.08	0.22 ± 0.15	0.18 ± 0.07	0.704
C16:0	9.50 ± 3.54	11.38 ± 5.19	8.13 ± 3.11	0.293
C17:0	0.43 ± 0.17	0.49 ± 0.22	0.38 ± 0.13	0.463
C18:0	6.59 ± 2.25	7.54 ± 3.11	6.30 ± 1.85	0.582
C20:0	0.06 ± 0.02	0.07 ± 0.03	0.06 ± 0.02	0.416
C21:0	0.05 ± 0.01	0.06 ± 0.03	0.05 ± 0.02	0.458
C23:0	0.07 ± 0.02	0.08 ± 0.02	0.09 ± 0.02	0.068
C14:1n-5	0.05 ± 0.03	0.08 ± 0.05	0.04 ± 0.02	0.194
C16:1n-7	0.79 ± 0.29	1.02 ± 0.52	0.74 ± 0.31	0.317
C17:1n-8	0.25 ± 0.02	0.26 ± 0.02	0.27 ± 0.04	0.370
9t-C18:1	0.20 ± 0.06	0.24 ± 0.10	0.19 ± 0.07	0.400
C18:1n-7	0.72 ± 0.29	1.00 ± 0.55	1.02 ± 0.40	0.319
C18:1n-9	16.22 ± 5.30	18.87 ± 7.68	14.41 ± 4.70	0.352
C20:1n-9	0.58 ± 0.07	0.59 ± 0.13	0.70 ± 0.14	0.073
C22:1n-9	0.15 ± 0.04	0.15 ± 0.03	0.15 ± 0.06	0.975
9t,12t-C18:2	0.13 ± 0.05	0.15 ± 0.06	0.12 ± 0.04	0.539
C18:2 n-6	2.23 ± 0.17	2.26 ± 0.33	2.28 ± 0.26	0.942
C18:3 n-3	0.59 ± 0.07	0.60 ± 0.13	0.72 ± 0.14	0.073
C20:2n-6	0.03 ± 0.00	0.03 ± 0.01	0.02 ± 0.00	0.230
C20:3n-9	0.26 ± 0.02	0.26 ± 0.04	0.30 ± 0.05	0.072
C20:3n-6	0.11 ± 0.01	0.10 ± 0.01	0.11 ± 0.01	0.358
C20:3n-3	0.01 ± 0.00	0.01 ± 0.00	0.01 ± 0.01	0.975
C20:4n-6	0.99 ± 0.11 ^b^	0.92 ± 0.04 ^b^	1.13 ± 0.10 ^a^	<0.001
C20:5n-3	0.66 ± 0.09	0.63 ± 0.06	0.73 ± 0.10	0.080
C22:6n-3	0.29 ± 0.07	0.25 ± 0.03	0.27 ± 0.05	0.309
SFA	18.21 ± 6.69	21.59 ± 9.80	16.25 ± 5.69	0.381
MUFA	18.95 ± 6.03	22.20 ± 9.05	17.52 ± 5.61	0.414
PUFA	5.30 ± 0.41	5.21 ± 0.61	5.68 ± 0.46	0.164
Total fat ^4^	46.35 ± 14.34	53.49 ± 21.24	43.07 ± 12.83	0.965

^a,b^ Means in the same rows differ significantly (*p* < 0.05);.^1^ Control diet = CD, Seaweed supplemented diet = SD, Pasture diet = PD; ^2^ Ultimate pH measured in *Semimembranosus* muscle; ^3^ Heme content and fatty acids measured in raw meat (*Semimembranosus* + *Adductor*); ^4^ Total fat = Total fatty acids/0.916 [16].

**Table 3 foods-11-01043-t003:** Micronutrient content in raw meat and dry-cured leg from lamb fed three finishing diets. Results are presented as mean ± STD (*n* = 8).

	CD ^1^		SD		PD		*p*-Value
	Raw Meat	Dry-Cured Leg	Raw Meat	Dry-Cured Leg	Raw Meat	Dry-Cured Leg	
Selenium, Se ^2^	15.88 ± 1.02 ^b^ (14.80–18.17) ^3^	15.54 ±1.03 ^a^ (14.17–17.72)	18.20 ± 0.88 ^a,x^ (16.45–19.32)	16.88 ± 0.53 ^a,y^ (16.25–17.62)	11.97 ± 2.0 ^c^ (9.15–14.45)	12.09 ± 1.89 ^b^ (9.23–13.95)	<0.001
Iodine, I	<1.9 ^b^ <1.9	2.56 ± 0.51 ^b^ (2.51–4.82)	55.69 ± 12.62 ^a,x^ (47.61–85.44)	95.51 ± 16.59 ^a,y^ (90.11–161.87)	<1.9 ^b^ <1.9	1.43 ± 0.09 ^b^ (1.90–2.24)	<0.001
Arsenic, As	0.11 ± 0.05 ^b^ (0.07–0.23)	0.14 ± 0.02 ^b^ (0.12–0.16)	3.79 ± 0.39 ^a,x^ (3.14–4.33)	4.47 ± 0.80 ^a,y^ (3.54–5.74)	0.13 ± 0.11 ^b^ (0.06–0.35)	0.10 ± 0.02 ^b^ (0.06–0.13)	<0.001

^a–c^ Different subscripts in the same raw indicate differences between dietary treatment; ^x,y^ Different subscripts in the same raw indicate differences between raw meat and dry-cured leg within the same dietary treatment;. ^1^ Control diet = CD, Seaweed supplemented diet = SD, Pasture diet = PD; ^2^ Mineral content was expressed as µg/100 g of raw meat or dry-cured leg; ^3^ Min-Max content of micronutrients.

**Table 5 foods-11-01043-t005:** Metabolite profile of dry-cured leg and variations of the metabolites between the three lamb finishing diets (Log2 Fold change values, Log2 FC).

						Log2 FC		
Compound	RT	RI	CD	SD	PD	CD: SD ^1^	CD: PD	SD: PD
glycine	13.16	1045.72	4.85 × 10^9^	5.02 × 10^9^	2.98 × 10^9^	−1.20	0.54	1.74
leucine	14.36	1083.97	3.24 × 10^9^	5.85 × 10^9^	3.88 × 10^9^	−1.19	1.14	2.33
valine	16.17	1023.17	1.05 × 10^10^	1.13 × 10^10^	9.57 × 10^9^	−0.45	0.56	1.00
isoleucine	18.64	1066.58	6.20 × 10^9^	9.03 × 10^9^	6.71 × 10^9^	−1.17	−0.11	1.05
proline	18.79	1069.23	2.87 × 10^9^	2.90 × 10^9^	1.76 × 10^9^	−0.35	0.84	1.19
β-alanine	22.48	1075.85	4.83 × 10^8^	9.38 × 10^8^	4.94 × 10^8^	−1.15	1.17	2.32
serine	20.57	1001.36	1.50 × 10^10^	1.31 × 10^10^	9.24 × 10^9^	−0.06	2.98	3.04
aspartic acid	24.95	1076.40	4.80 × 10^9^	2.92 × 10^9^	2.95 × 10^9^	0.54	1.50	0.95
methionine	24.97	1077.06	2.69 × 10^9^	3.54 × 10^9^	3.05 × 10^9^	−1.09	−0.08	1.01
glutamic acid	27.55	1088.53	1.98 × 10^10^	2.21 × 10^10^	2.04 × 10^10^	−0.70	0.31	1.01
phenylalanine	27.75	1097.59	1.18 × 10^10^	1.08 × 10^10^	7.81 × 10^9^	−0.32	2.65	2.97
asparagine	28.75	1043.25	3.18 × 10^9^	1.93 × 10^9^	1.83 × 10^9^	0.37	1.99	1.62
glutamine	31.17	955.20	2.30 × 10^9^	2.39 × 10^9^	1.66 × 10^9^	−0.82	0.93	1.75
histidine	34.42	1120.40	3.52 × 10^9^	2.18 × 10^9^	2.61 × 10^9^	1.49	0.87	−0.62
tyrosine	34.83	1141.39	8.59 × 10^9^	8.00 × 10^9^	7.56 × 10^9^	−0.39	1.14	1.52
lactic acid	11.16	1082.23	8.84 × 10^10^	1.04 × 10^11^	8.48 × 10^10^	−1.87	−1.31	0.56
glycolic acid	11.59	1095.48	1.14 × 10^8^	2.00 × 10^8^	1.32 × 10^8^	−0.84	0.74	1.57
4-hydroxybutanoic acid	16.88	1035.71	1.02 × 10^8^	2.22 × 10^8^	1.10 × 10^8^	−1.60	1.27	2.86
butanedioic acid	19.32	1078.56	3.23 × 10^9^	5.09 × 10^9^	4.06 × 10^9^	−2.02	−0.32	1.71
2-aminomalonic acid	23.55	1018.29	4.01 × 10^8^	7.95 × 10^8^	7.76 × 10^8^	−1.53	−0.71	0.82
malic acid	24.11	1041.62	3.69 × 10^8^	6.04 × 10^8^	6.91 × 10^8^	−1.10	0.30	1.41
gluconic acid	35.85	1193.17	8.25 × 10^8^	1.72 × 10^9^	7.09 × 10^8^	−0.57	1.76	2.32
9H-purin-6-ol	32.06	1000.51	1.33 × 10^9^	1.65 × 10^9^	1.77 × 10^9^	−0.60	0.84	1.44
scyllo-inositol	36.44	1223.20	4.92 × 10^8^	7.91 × 10^8^	7.45 × 10^8^	−0.91	0.54	1.45
myo-inositol	37.74	1289.08	1.07 × 10^10^	1.38 × 10^10^	1.38 × 10^10^	−0.38	0.99	1.36
fructopyranose	31.91	992.87	1.89 × 10^9^	2.12 × 10^9^	1.51 × 10^9^	−0.44	1.86	2.30
fructose	33.17	1056.73	7.53 × 10^9^	1.04 × 10^10^	5.67 × 10^9^	−1.47	1.04	2.51
sorbose	33.38	1067.80	6.85 × 10^9^	6.97 × 10^9^	4.12 × 10^9^	−0.52	1.73	2.25
glucose	33.49	1073.18	1.01 × 10^10^	1.16 × 10^10^	9.85 × 10^9^	−0.23	1.94	2.17
glucopyranose	33.68	1083.04	1.62 × 10^10^	2.19 × 10^10^	1.42 × 10^10^	−0.60	4.23	4.83
mannose	33.76	1086.80	3.58 × 10^10^	3.79 × 10^10^	2.45 × 10^10^	−0.22	4.64	4.86
palmitic acid	37.11	1256.98	2.98 × 10^9^	2.76 × 10^9^	3.20 × 10^9^	−1.46	0.16	1.62
9,12-octadecadienoic acid (Z,Z)- (18:2)	40.31	1419.39	3.66 × 10^8^	3.58 × 10^8^	1.15 × 10^9^	−4.86	−3.15	1.71
oleic acid	40.44	1425.90	5.37 × 10^9^	4.70 × 10^9^	6.85 × 10^9^	−0.73	0.65	1.38
stearic acid	40.89	1448.86	1.50 × 10^9^	1.37 × 10^9^	2.13 × 10^9^	−0.52	0.52	1.03
arachidonic acid	43.14	1563.03	3.10 × 10^8^	4.28 × 10^8^	5.14 × 10^8^	−1.58	−0.32	1.26
eicosapentaenoic acid	43.20	1566.33	3.49 × 10^8^	4.40 × 10^8^	7.10 × 10^8^	−1.16	−0.98	0.18
urea	17.70	1050.09	6.51 × 10^9^	1.02 × 10^10^	8.07 × 10^9^	−1.18	0.89	2.07
ethanolamine	17.72	1050.39	2.65 × 10^9^	1.83 × 10^9^	1.33 × 10^9^	−3.13	−6.50	−3.36
niacinamide	24.13	1042.57	2.19 × 10^8^	3.70 × 10^8^	3.11 × 10^8^	−1.20	0.61	1.81
pyroglutamic acid	25.05	1080.47	9.50 × 10^9^	1.41 × 10^10^	8.64 × 10^9^	−1.06	0.60	1.66
creatinine	25.83	1013.43	2.12 × 10^9^	3.04 × 10^9^	1.13 × 10^9^	−0.76	1.86	2.62
ornithine	30.68	965.94	7.08 × 10^8^	3.86 × 10^8^	6.78 × 10^8^	−0.81	0.38	1.19
norvaline	37.20	1261.81	3.58 × 10^8^	4.88 × 10^8^	3.35 × 10^8^	−1.25	0.08	1.33
inosine	46.43	1730.42	1.88 × 10^10^	1.68 × 10^10^	1.83 × 10^10^	−0.13	1.07	1.20
cholesterol	55.37	2184.21	5.22 × 10^8^	5.54 × 10^8^	6.68 × 10^8^	−1.25	−0.11	1.14

^1^ Log2 FC Fold change for metabolites that were downregulated (green) or upregulated (red) (FC>2) in dry-cured leg from control vs. seaweed supplemented (CD:SD), control vs pasture (CD:PD), seaweed supplemented vs. pasture (SD:PD).

## Data Availability

Data is contained within the article.

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
