# Peer review of "Seaweed Inclusion in Finishing Lamb Diet Promotes Changes in Micronutrient Content and Flavour-Related Compounds of Raw Meat and Dry-Cured Leg (Fenalår)"

_foods, 2022, doi:10.3390/foods11071043_

Round 1
Reviewer 1 Report
The objective of the present work was to investigate the effect of finishing lamb diet supplemented with seaweed on the quality of raw meat and deboned dry-cured lamb leg. The experiment is well designed, the discussion is consistent, and the final conclusions are interesting
Some aspects of the manuscript should be improved or revised
Line 26. Please provide a general conclusion at the end of the abstract
Line 62. Animals selection should be Animals Selection (uppercase)
Line 63. Please include the animal age
Line 79. Raw material should be Raw Material (uppercase)
Line 107. Production of dry-cured leg should be Production of Dry-cured Leg (uppercase)
Lines 127. Instrumental colour should be Instrumental Colour (uppercase)
Line 134. Chemical analysis should be Chemical Analysis (uppercase)
Line 170. Volatile compound analysis should be Volatile Compound Analysis (uppercase)
Line 200. Metabolite analysis should be Metabolite Analysis (uppercase)
Line 225. Sensory analysis should be Sensory Analysis (uppercase)
Lines 241. Statistical analysis should be Statistical Analysis (uppercase)
Line 265. Quality properties should be Quality Properties (uppercase)
Line 273. “Similar results for were reported…” delete “for”
Line 274. Include the legend “Data reported are mean values together with standard deviation (n = ?)” include n value
Line 274. Data in Table 2. Separate numbers from symbol “±”
Line 283. Micronutrient content should be Micronutrient Content (uppercase)
Line 319. Include the legend “Data reported are mean values together with standard deviation (n = ?)” include n value
Line 319. Data in Table 3. Separate numbers from symbol “±”
Line 326. Volatile compounds should be Volatile Compounds (uppercase)
Line 368. Include the legend “Data reported are mean values together with standard deviation (n = ?)” include n value
Line 368. Data in Table 4. Separate numbers from symbol “±”
Line 368. Table 4. Arrange rows and columns properly
Line 378. Metabolite changes should be Metabolite Changes (uppercase)
Line 423. Separate Table 6 from Table 5
Line 423. Data in Table 6. Separate numbers from symbol “±”
Line 424. Include the legend “Data reported are mean values together with standard deviation (n = ?)” include n value
Line 430. Sensory profile of dry-cured leg should be Sensory Profile of Dry-cured Leg (uppercase)
Line 456. “B” letter is not specified in Figure 1
Line 477. Journal name in all references should be in italic
Line 477. Journal year in all references should be in bold
Author Response
Detailed response to Reviewer #1:
The objective of the present work was to investigate the effect of finishing lamb diet supplemented with seaweed on the quality of raw meat and deboned dry-cured lamb leg. The experiment is well designed, the discussion is consistent, and the final conclusions are interesting.
Response:
The authors thank to Reviewer #1 for a nice comments.
Reviewer #1:
Some aspects of the manuscript should be improved or revised.
Response:
The authors revised Manuscript carefully taking into consideration all specific comments provided by Reviewer #1.
- Line 26. Please provide a general conclusion at the end of the abstract
Response:
In agreement with the Reviewer’s comment, the abstract section was amended: This study showed the potential of seaweed in iodine biofortification of lamb meat and dry-cured products. Iodine-rich meat products should reduce iodine-deficiency among humans.
- Line 62. Animals selectionshould be Animals Selection(uppercase)
Response:
The change was done.
- Line 63. Please include the animal age
Response:
In lines 104-109 was added information about animal age: Twenty-four Norwegian White female lambs (37.3 ± 1.6 kg BW) about 6 months old were randomly divided in three groups and fed ad-libitum with finishing diets for 35 days: 1) control diet (CD) ‒ total mixed ration of grass silage totally mixed with compound feed, rolled barley and mineral premix, 2) seaweed supplemented diet (SD) ‒ 5% of the control diet was replaced with dried Saccharina latissima on DM basis, and 3) permanent ley pasture (PD) ‒ for the whole experimental period animals were kept on pasture.
- Line 79. Raw materialshould be Raw Material(uppercase)
Response:
The change was done.
- Line 107. Production of dry-cured legshould be Production of Dry-cured Leg(uppercase)
Response:
The change was done.
- Lines 127. Instrumental colourshould be Instrumental Colour (uppercase)
Response:
The change was done.
- Line 134. Chemical analysisshould be Chemical Analysis(uppercase)
Response:
The change was done.
- Line 170. Volatile compound analysisshould be Volatile Compound Analysis(uppercase)
Response:
The change was done.
- Line 200. Metabolite analysisshould be Metabolite Analysis(uppercase)
Response:
The change was done.
- Line 225. Sensory analysisshould be Sensory Analysis(uppercase)
Response:
The change was done.
- Lines 241. Statistical analysisshould be Statistical Analysis(uppercase)
Response:
The change was done.
- Line 265. Quality propertiesshould be Quality Properties(uppercase)
Response:
The change was done.
- Line 273. “Similar results for were reported…” delete “for”
Response:
The change was done.
- Line 274. Include the legend “Data reported are mean values together with standard deviation (n = ?)”include n value
Response:
The legend of Table 2 was revised: Table 2. pH, heme content, and fatty acid profile of raw meat from lamb fed three finishing diets. Results are presented as mean ± STD (n = 8).
- Line 274. Data in Table 2. Separate numbers from symbol “±”
The changes were done.
- Line 283. Micronutrient contentshould be Micronutrient Content(uppercase)
Response:
The change was done.
- Line 319. Include the legend “Data reported are mean values together with standard deviation (n = ?)” include n value
Response:
The legend of Table 3 was revised: Table 3. Micronutrient content in raw meat and dry-cured leg from lamb fed three finishing diets. Results are presented as mean ± STD (n = 8).
- Line 319. Data in Table 3. Separate numbers from symbol “±”
Response:
The changes were done in Table 3.
- Line 326. Volatile compoundsshould be Volatile Compounds(uppercase)
Response:
The change was done.
- Line 368. Include the legend “Data reported are mean values together with standard deviation (n = ?)” include n value
Response:
The legend of Table 4 was revised: Table 4. Differences in volatile compounds (p < 0.05) between raw meat and dry-cured leg from lamb fed three finishing diets. Results are presented as mean ± STD (n = 8).
- Line 368. Data in Table 4. Separate numbers from symbol “±”
Response:
The changes were done in Table 4.
- Line 368. Table 4. Arrange rows and columns properly
Response:
Table 4 was arranged in agreement with the Reviewer’s comment.
- Line 378. Metabolite changesshould be Metabolite Changes(uppercase)
Response:
The change was done.
- Line 423. Separate Table 6 from Table 5.
Response:
Table 5 was moved upper, at the beginning of the Metabolite Changes section.
- Line 423. Data in Table 6. Separate numbers from symbol “±”
Response:
The change was done.
- Line 424. Include the legend “Data reported are mean values together with standard deviation (n = ?)” include n value
Response:
The legend of Table 6 was revised: Table 6. Differences in metabolites (p < 0.05) between dry-cured leg produced from lamb fed three finishing diets. Results are presented as mean ± STD (n = 8).
- Line 430. Sensory profile of dry-cured legshould be SensoryProfile of Dry-cured Leg (uppercase)
Response:
The changes were done.
- Line 456. “B” letter is not specified in Figure 1
Response:
The missing identification (B) in Figure 1 was added.
- Line 477. Journal name in all references should be in italic
Response:
The changes were done.
- Line 477. Journal year in all references should be in bold.
Response:
The changes were done.

Reviewer 2 Report
The results in this paper give a better understanding of the finishing lamb diet and flavour-related compounds. I would like to accept after minor revision. Further investigation will be needed in the overall odor and taste sensory analysis.
Line 7-10: Please write in detail the organization to which the authors belong.
(IRTA, Catalonia Spain, Nortura SA)
Line 135-136: Please explain the extraction of fatty acid in detail, i.e., the volume of 10 N KOH in water and MeOH.
Line 201-202: Was it “homogenized” dry-cured leg?
Line 212: Why did you use “thus”? Is it not “thus” but “then”?
Line 227: Please describe the semi-trained panel in detail. Have any references?
Table 2: Are values represented as the mean ± standard deviation? It seems high standard deviations in fatty acids, especially C16:0 and C18:1n-9. Is there any problem with the fatty acid analysis method?
Table 3: It also seems high standard deviations in micronutrient, especially Iodine. Is there any problem with the analysis method?
Table 3 and 4: Please correct the table to make it easier to see. Regarding Table 4, how about writing meat and dry-cured meat in two lines? I would think “P” and “D × P” are not necessary, how about it?
Line422: control vs seaweed supplemented (CD : SD)
Figure 1: There is no "B" notation. In addition, Items that are statistically different (bite through, tenderness, and softness) should be marked in the figure.
Line448-449: Why had the CD dry-cured leg significantly higher tenderness (p = 0.007) and softness (p = 0.014) compared with SD and PD? Do you have the moisture content data concerned with the texture profile?
Author Response
Detailed response to Reviewer #2:
The results in this paper give a better understanding of the finishing lamb diet and flavour-related compounds. I would like to accept after minor revision. Further investigation will be needed in the overall odor and taste sensory analysis.
Response:
Taking into consideration the comment of Reviewer #2, the authors have done improvements in the Manuscript.
- Line 7-10: Please write in detail the organization to which the authors belong.
(IRTA, Catalonia Spain, Nortura SA)
Response:
The improvements were done in agreement with the Reviewer’s comment:
1 Faculty of Chemistry, Biotechnology and Food Science, Norwegian University of Life Sciences, NO-1430 Ås, Norway; bjorg.egelandsdal@nmbu.no (B.E.)
2 IRTA, Food Technology and Food Safety Programs, Finca Camps i Armet, E-17121 Monells, Girona, Spain; ecollbrasas@gmail.com (E.C-B.); elena.fulladosa@irta.cat (E.F.)
3 Nortura SA, P.O. Box 360 Økern, NO-0513 Oslo, Norway; elin.hallenstvedt@nortura.no (E.H.); per.berg@nortura.no (P.B.)
4 Animalia, P.O. Box 396 Økern, NO-0513 Oslo, Norway; torunn.haseth@animalia.no (T.T.H.)
5 Faculty of Bioscience, Norwegian University of Life Sciences, P.O. Box 5003, NO-1432 Ås, Norway; margareth.overland@nmbu.no (M.Ø.)
*Correspondence: vladana.grabez@nmbu.no; Tel.: +47-46523143
- Line 135-136: Please explain the extraction of fatty acid in detail, i.e., the volume of 10 N KOH in water and MeOH.
Response:
In agreement with the comment, FAME extraction is explained in detail: Extraction of fatty acid methyl esters (FAME) from 0.25 g homogenized raw meat was performed as described by Yi et al. [10]. Briefly, homogenized meat sample was mixed with 1 mL of tridecanoic acid (0.5 mg C13:0/mL methanol) as an internal standard. Then, the sample was dissolved and hydrolyzed with 0.56 mL of 10 N KOH in water and 4.2 mL of methanol. The incubation of the samples in water bath at 55 ℃ for 1.5 h with hand shaking for 5 s every 20 min was performed. The samples were cooled, 0.46 mL of 24 N sulphuric acid in water was added, then once again incubated and cooled as previously described. Mixing with 3 mL of hexane for 5 min and centrifuging at 653 × g for 10 min the fatty acid methyl esters (FAME) were separated. The hexane layer with FAME was transferred to GC vials and kept at -20 ºC prior to the analysis.
- Line 201-202: Was it “homogenized” dry-cured leg?
Response:
Metabolite extraction, derivatization and analysis were performed on homogenized dry-cured leg [14].
- Line 212: Why did you use “thus”? Is it not “thus” but “then”?
Response:
Deconvolution and integration of peaks were performed with MassHunter Qual (version B.07.00, Agilent Technologies, Santa Clara, CA, USA), and identification using NIST17 (National Institute of Standards and Technology, Gaithersburg, MD, USA) with ≥ 70% mass spectral match.
- Line 227: Please describe the semi-trained panel in detail. Have any references?
Response: The sensory analyses were performed by internal semi-trained panel that is reliable and trained to evaluate quality properties of dry-cured products. The selected panellists have strong interest into food having background as chefs or butchers. In addition, panellists are trained to define sensory profile and taste sensitivity.
We added: The internal semi-trained panel is reliable and trained to evaluate sensory properties of dry-cured lamb products. The selected panellists are yearly re-trained for sensitivity of recognition thresholds of basic tastes (sweet, sour, salty, bitter, and umami) and rancidity.
- Table 2: Are values represented as the mean ± standard deviation? It seems high standard deviations in fatty acids, especially C16:0 and C18:1n-9. Is there any problem with the fatty acid analysis method?
Response:
An internal standard C13:0 was used to inspect the variations that occurred during fatty acid (FA) analysis on GC-FID and at average deviation was around 6% for ISTD. The FA content is expressed at mean ± std. In addition, for these analysis the homogenate (Semimembranosus + Adductor) was used and it could have an impact on FA differences between biological samples/ individual animals.
- Table 3: It also seems high standard deviations in micronutrient, especially Iodine. Is there any problem with the analysis method?
Response:
Iodine analysis were performed in the accredited laboratory of National Institute of Health Doutor Ricardo Jorge (INSA, Lisboa, Portugal). In their report, for lamb meat the deviation between three replicates was 5.3% (https://www.euro.who.int/en/countries/portugal/publications/scientific-update-on-the-iodine-content-of-portuguese-foods-2018). In addition, authors have obtained iodine data from different laboratories and experiments with inclusion of seaweed in ruminant diet and identified similar variations between biological samples as in the present study. Therefore, authors are confident that large variations in micronutrient contents in the present Manuscript are not result of the analytical methods.
- Table 3 and 4: Please correct the table to make it easier to see. Regarding Table 4, how about writing meat and dry-cured meat in two lines? I would think “P” and “D × P” are not necessary, how about it?
Response:
In agreement with Reviewer’s comment, the authors improved Table 3 layout, whereas Table 4 is kept in the original form. It’s of specific relevance to give an insight into complexity of flavour development during processing from raw meat to dry-cured product, therefore authors would rather keep “P” and “D × P” effects.
- Line422: control vs seaweed supplemented (CD : SD)
Response:
The corrections were done: 1 Log2 FC Fold change for metabolites that were downregulated (green) or upregulated (red) (FC>2) in dry-cured leg from control vs seaweed supplemented (CD : SD), control vs pasture (CD : PD), seaweed supplemented vs pasture (SD : PD).
- Figure 1: There is no "B" notation. In addition, Items that are statistically different (bite through, tenderness, and softness) should be marked in the figure.
Response:
In agreement with Reviewer’s comments Figure 1 was amended. Missing identification (B) and the significance for texture attributes between the diets was added.
- Line 448-449: Why had the CD dry-cured leg significantly higher tenderness (p = 0.007) and softness (p = 0.014) compared with SD and PD? Do you have the moisture content data concerned with the texture profile?
Response: The results obtained for tenderness could be related with up-regulation of free amino acids in CD and SD dry-cured leg as shown in Table 6.